# Suitability of IgG responses to multiple *Plasmodium falciparum* antigens as markers of transmission intensity and pattern

Eric Kyei-Baafour[1,2], Mavis Oppong[3], Kwadwo Asamoah Kusi[1], Abena Fremaah Frempong[1], Belinda Aculley[1], Fareed K. N. Arthur[2], Regis Wendpayangde Tiendrebeogo[4], Susheel K. Singh[4,5], Michael Theisen[4,5], Margaret Kweku[3], Bright Adu[1], Lars Hviid[4], Michael Fokuo Ofori[1]*

**1** Department of Immunology, Noguchi Memorial Institute for Medical Research, College of Health Sciences, University of Ghana, Accra, Ghana, **2** Department of Biochemistry and Biotechnology, College of Science, Kwame Nkrumah University of Science and Technology, Kumasi, Ghana, **3** Department of Epidemiology and Biostatistics, School of Public Health, University of Health and Allied Sciences, Hohoe, Ghana, **4** Centre for Medical Parasitology at Department of Immunology and Microbiology, University of Copenhagen, and at Department of Infectious Diseases, Rigshospitalet, Copenhagen, Denmark, **5** Department for Congenital Disorders, Statens Serum Institut, Copenhagen, Denmark

* mofori@noguchi.ug.edu.gh

**Data Availability Statement:** Data are available from figshare (DOI: 10.6084/m9.figshare. 14046911.v1).

## Abstract

Detection of antibody reactivity to appropriate, specific parasite antigens may constitute a sensitive and cost-effective alternative to current tools to monitor malaria transmission across different endemicity settings. This study aimed to determine the suitability of IgG responses to a number of *P. falciparum* antigens as markers of transmission intensity and pattern. Antibody responses to multiple malaria antigens were determined in 905 participants aged 1–12 years from three districts with low (Keta), medium (Hohoe) and high (Krachi) transmission intensity in the Volta region of Ghana. Blood film microscopy slides and dry blood spots (DBS) were obtained for parasitaemia detection and antibody measurement, respectively. Sera were eluted from DBS and levels of IgG specific for 10 malaria antigens determined by a multiplex assay. Results were compared within and among the districts. Total IgG responses to MSPDBL1, MSPDBL$_{Leucine}$, MSP2-$_{FC27}$, RAMA, and *Pf*Rh2a and *Pf*Rh2b were higher in Krachi than in Hohoe and Keta. Seroprevalence of IgG specific for MSPDBL$_{Leucine}$, RON4, and *Pf*Rh2b were also highest in Krachi. Responses to RALP-1, *Pf*Rh2a and *Pf*Rh2b were associated with patent but asymptomatic parasitaemia in Keta, while responses to MSPDBL1, MSPDBL$_{Leucine}$, MSP2-$_{FC27}$, RAMA, Rh2-$_{2030}$, and *Pf*Rh2b were associated with parasite carriage in Hohoe, but not in Krachi. Using ROC analysis, only *Pf*Rh2b was found to predict patent, but asymptomatic, parasitaemia in Keta and Hohoe. Antibody breadth correlated positively with age (r = 0.29, p<0.0001) and parasitaemia (β = 3.91; CI = 1.53 to 6.29), and medium to high transmission (p<0.0001). Our findings suggest differences in malaria-specific antibody responses across the three transmission zones and that *Pf*Rh2b has potential as a marker of malaria transmission intensity and pattern. This could have implications for malaria control programs and vaccine trials.

**Funding:** This study was supported by the Danish Research Council for Development Research (Grant No. 17-02-KU) awarded to LH and MFO. BAdu and MT are supported by Ministry of Foreign Affairs of Denmark (DFC file no.14-P01-GHA) and administered by DANIDA Fellowship Centre. The funders had no role in study design, data collection and interpretation, or the decision to submit the work for publication.

**Competing interests:** The authors have declared that no competing interests exist.

## Introduction

Malaria continues as a serious challenge to health systems in sub-Saharan Africa, despite increased efforts to control the disease. In Africa, 96% of malaria cases were due to *Plasmodium falciparum* in 2019. Out of the over 409,000 global deaths from malaria in 2019, 94% occurred in Africa [1].

In Ghana, malaria is endemic with the entire population at risk, and the disease accounts for about 30% of all out-patients [2]. Malaria transmission in Ghana differs among its three major ecological zones. It is lowest in the coastal shrub zone of southern Ghana, intermediate in the middle belt dominated by semi-deciduous and transitional forest, and highest in the northern part of the country, characterized by guinea savannah [3].

Malaria transmission intensity is measured using *Plasmodium* parasite prevalence i.e., the proportion of the population infected with the parasites. However, parasite prevalence is highly dependent on the method used to detect parasites in the blood of infected individuals [4]. Entomological inoculation rate (EIR) is another malaria transmission intensity indicator that shows the rate at which individuals are bitten by infective mosquitoes [5]. The estimation of transmission by EIR suffers from low precision as a result of temporal distributions of vectors [5,6] and from being labour-intensive [7]. Antibody responses to malaria-specific antigens have been suggested as alternative markers of malaria transmission intensity [8] and differences in transmission patterns [9,10]. Spatial heterogeneity in malaria transmission has therefore been estimated using serological tools [11–13].

Malaria-specific antibodies elicited by natural infection are generally considered markers of parasite exposure and good for sero-surveillance. These antibodies can also be used to predict parasite exposure over time [4]. However, some may not be able to define properly heterogeneity in malaria transmission, because of their persistence in circulation. In addition, estimating malaria transmission reliably with methods such as EIR and microscopy are becoming increasingly difficult as the prevalence of clinical cases declines. Changes in the burden of malaria in low-transmission settings may thus not be detected [14]. There is therefore a need to characterize parasite-specific immune responses in different transmission settings to select good markers for transmission monitoring. Furthermore, the characterization of antibody responses will enhance efforts to develop more accurate tools to monitor transmission [4,9,15].

Most serology studies of malaria transmission patterns have focused on few antigens such as circumsporozoite protein (CSP), cell-traversal protein for ookinetes and sporozoites (CelTOS), apical membrane antigen 1 (AMA1), and merozoite surface protein 1 (MSP1) [8,9,16,17]. However, many other antigens need evaluation to expand the repertoire used to determine heterogeneities in malaria transmission. Ten antigens were selected for this study: merozoite surface protein Duffy binding ligand 1 (MSPDBL1, MSPDBL-$_{Leucine}$), erythrocyte-binding antigen (EBA140RIII-V), merozoite surface protein 2 (MSP2$_{FC27}$), rhoptry-associated like protein (RALP-1), rhoptry-associated membrane antigen RAMA, *Plasmodium falciparum* reticulocyte homologue (*Pf*Rh2$_{-2030}$, *Pf*Rh2A, *Pf*Rh2B), and rhoptry-associated neck protein (RON4). All are bloodstage antigens, and some have been found to be associated with protection from malaria and to be good markers of exposure [18,19].

Malaria transmission studies in the eastern part of Ghana are scanty and no study has compared anti-malarial antibody responses across the three ecological zones spanned by the Volta region in Ghana. Entomological inoculation rates increase from the southern coastal belt (62.1 infective bites/person/year) [20], through the middle forest transition zone (269 infective bites/person/year) [21], to the guinea savannah zone in the north (643 infective bites/person/year) [22]. In this study, we compared antibody levels to multiple malaria antigens in three

districts, each represents one of the three ecological zones in the Volta region of Ghana, to assess their suitability to determine their suitability as markers of transmission intensity and pattern.

## Material and methods

### Study site and population

This cross-sectional community-based survey was carried out in December 2018 in three randomly selected districts in the Volta region, each with a different ecological setting. The study involved children aged 1–12 years living in communities within the selected districts. The study site and population have been described elsewhere [23]. Briefly, the study employed a multi-stage sampling method in which the three [3] ecological zones of the Volta Region, southern, middle and northern zones reflecting the three malaria transmission zones, were used. This approach was used to ensure that the results obtained could reflect the entire population of Volta Region. One district each from the three zones was randomly picked. The communities in the selected districts were also listed and picked randomly. Children were only included in the study after informed consent has been granted by parents/guardians, and were permanently resident in the selected districts. Febrile children and those who were reported ill two weeks before sampling by parents/guardians were excluded from the study. Children with congenital defects were also not included in the study.

The three selected districts were Krachi (guinea savannah zone) in the north, Hohoe (forest transition zone) in the middle, and Keta (coastal shrub and grassland zone) along the coast in the south. The Volta region, like other parts of Ghana, has two major seasons: a dry season from November to April, and a rainy season, which spans the months from May to October. However, rainfall in the areas around Hohoe and Keta is bimodal with a major peak in June and a minor peak in October [24]. Krachi and the adjoining districts experience one rainy season peaking in August [25]. Malaria prevalence in the Volta region by RDT and microscopy in 2014 were 36.8% and 25.2% respectively [26]. The locations of the three study sites are shown (Fig 1).

### Sampling

Ethical Approval for the study was obtained from the Research Ethics Committee of the University of Health and Allied Sciences (UHAS-REC A.1 (5) 18–19). Written informed consent was obtained from parents/guardians of children aged 1–12 years, and finger-prick blood spots (DBS) were collected using Whatman No. 5 filter paper (GE Healthcare, England). The spots were air-dried and stored desiccated at 4°C until ready for use. Thick and thin slides were also prepared for malaria microscopy.

**Examination of blood smears.** Giemsa-stained thick and thin blood smears were examined using a light microscope. A slide was considered negative if no parasites were observed after examining 200 microscopic fields at 100× magnification of the thick smear. Parasite density was estimated against 500 leucocytes, using an assumed leucocyte count of 8,000/μL of blood and expressed as parasites/μL. Polymerase Chain Reaction (PCR) data to measure submicroscopic parasitaemia from the same cohort has already been published elsewhere [23].

*Elution of serum from filter paper*. Elution of serum from the filter paper was done using a protocol described by Corran *et al.* [27] with slight modifications. Briefly, zip-lock bags containing individually wrapped filter papers were removed from storage and allowed to warm to room temperature for 30 minutes. Dried blood spots of about 2.5 mm diameter were cut using a leather punch into 96-well microtiter plates. To elute serum from spots, 150 μL PBS with 0.05% Tween 20, and 0.05% sodium azide was added to each well containing punch-outs, after

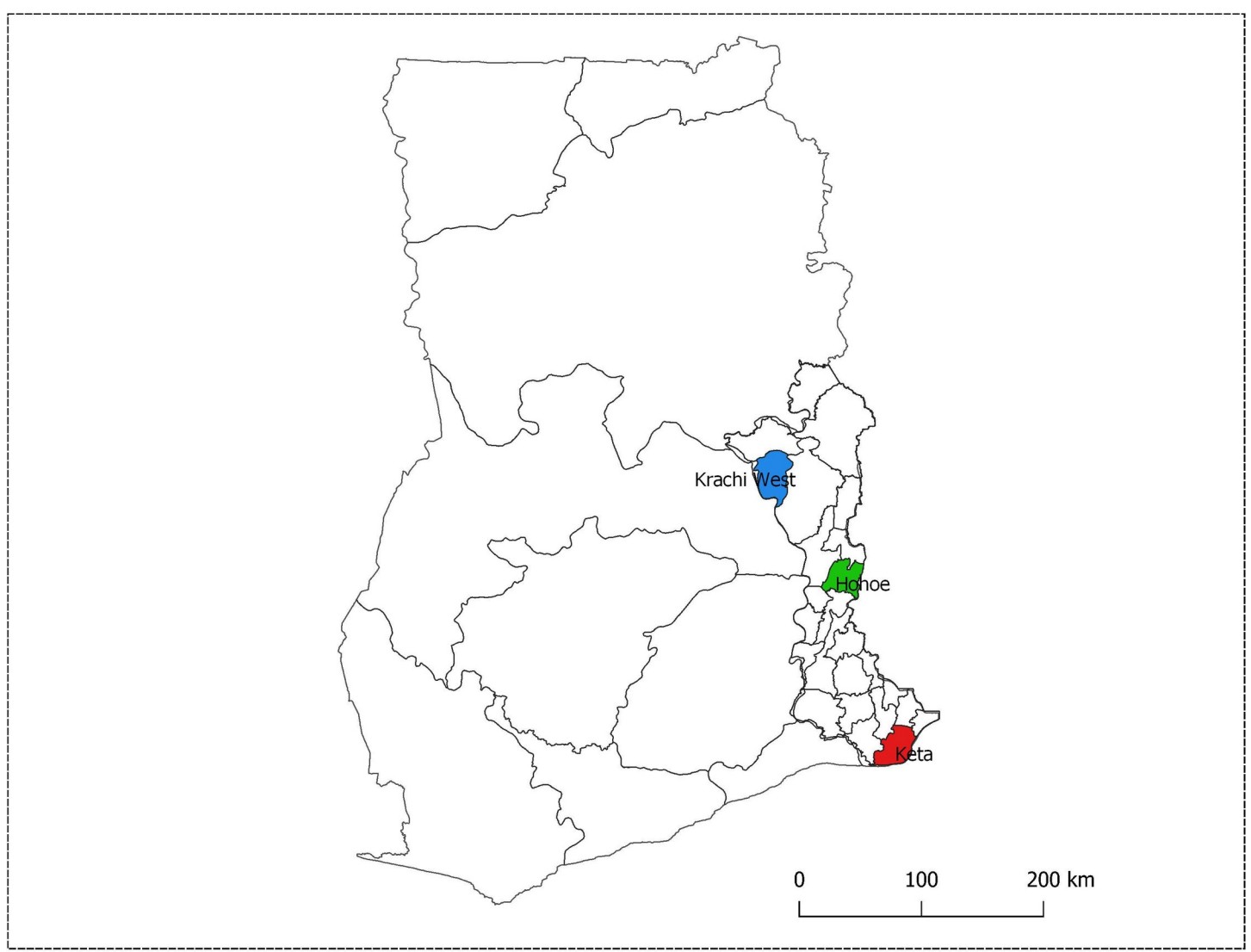

**Fig 1. Map of Ghana located in West Africa showing the region and districts where sampling was done.** This was the map of Ghana with the 10 regions as they existed before December 2019 when the study was performed. The red shaded area is Keta, the blue is Krachi and the green area is Hohoe. (Map was created using QGIS version 3.4.7 by Miss Jessica Asante, Department of Epidemiology, School of Public Health, University of Ghana).

which each plate was placed on a shaker overnight at 150 rpm. Finally, eluted samples were aliquoted into separate 96-well microtiter plate and kept at -20°C until use.

*Coupling of antigens to microsphere beads.* The antigens used here (MSPDBL1, EBA140R-III-V, MSPDBL$_{Leucine}$, MSP2$_{FC27}$, RALP-1, RAMA, *Pf*Rh2-$_{2030}$, *Pf*Rh2A, *Pf*Rh2B, and RON4), were expressed in *Lactococcus lactis* expression system using *P. falciparum* 3D7 variants with only MSP2$_{FC27}$ cloned from *P.falciparum* FC27 strain, as described previously [28]. These antigens were selected based on their different localization in/on merozoites [29], to determine their usefulness as transmission monitoring markers Antigen coupling was also done as previously described [30,31], with slight modifications. Briefly, each antigen was covalently coupled to microsphere beads with each bead region noted according to the manufacturer's protocol (Luminex). The regions used were MSPDBL1-{66}, EBA140RIII-V-{32}, MSPDBL$_{Leucine}$-{52}, MSP2$_{FC27}$-{35}, RALP-1-{45}, RAMA-{77}, *Pf*Rh2$_{-2030}$-{33}, *Pf*Rh2a-{47}, *Pf*Rh2b-{80}, and

RON4-{56}. BSA coupled to the bead region {89} was used as a control. Coupled beads were stored at 4˚C until use.

*Measurement of antigen-specific antibody levels by multiplex assay.* Antibodies with specificity for the panel of 10 *P. falciparum* recombinant antigens in eluted serum were measured on the Luminex 200 x-MAP platform (Luminex Inc., Austin, TX USA) as described previously [30], with slight modifications. Nine hundred and five (905) eluted samples were analysed. Briefly, a multiscreen filter base plate (Millipore, Billerica, MA) was pre-wetted with 100 μL/well freshly prepared assay buffer (PBS, 0.05% Tween 20, 1% BSA, 0.05% sodium azide, pH 7.4) for 30 minutes. Eluted sera and negative control sera from malaria naïve individuals (individuals with no travel history to any malaria-endemic country and deemed not to have encountered any malaria antigen) were diluted 1:5 giving a final dilution of 1:500. Adult plasma samples found to have higher responses to all the antigens included in the study were pooled and used as a positive control. Approximately 1,250 coupled beads from each region were mixed in equal volumes and added to each well at 50 μL/well of the pre-wetted plates. The plates were washed three times with assay buffer and test samples and controls added at 100 μL/well. Plates were incubated in the dark for 2 hours on a shaker at a speed of 300 rpm. After three washes, plates were incubated with biotin-conjugated goat-anti-human IgG (KPL, Gaithersburg, MD, USA) (diluted at 1:500, 100 μL/well) for 1 h. This was followed by incubation with 100 μL/well streptavidin-conjugated phycoerythrin (Thermo Fisher) diluted at 1:200 dilution for 30 minutes in the dark. After three washes, 100 μL assay buffer was added and plates subsequently read with the Luminex 200 system (Luminex Corporation). Results were expressed as mean fluorescence intensity (MFI).

*Data analysis.* Data was normalized for inter-plate and day-to-day variation by dividing the test sample on each plate by the positive control of the assay plate. This was then multiplied by the total mean positive control for all the plates to obtain the normalized MFI values using the formula:

(Sample/Plate Positive control) x Mean positive control for all plates)

Mean fluorescence intensity of samples obtained from malaria-naïve volunteers were used to define a cut-off for seropositivity of antibodies, calculated as the mean naïve MFI plus 2 standard deviations of the mean. Seroprevalence was antigen-specific. Differences in median antibody levels across the three districts were also determined using the Kruskal-Wallis test. Antibody levels were then $log_{10}$-transformed for further analysis. Age was stratified into two groups, those below 5 years and those above 5 years, and student's t-test used to determine differences in the $log_{10}$-transformed MFI in the two age groups for each district. Participants were divided into two subsets based on malaria positivity, and multiple logistic tests used to determine antibody responses associated with parasite carriage. Receiver operating characteristics (ROC) curves were fitted to predict parasite carriage using antibody levels as predictors. Antibody levels were categorized based on quartiles and assigned 0 for lowest, 1 for the second, 2 for third, and 3 for the highest quartile. The scores were summed up for the 10 antigens for each individual to generate breadth scores. The relationship between malaria transmission (defined using infective bites per person per year [20–22]) and breadth scores was determined using linear regression adjusting for age and bed net usage. Analysis and graphs were done using R statistical software version 4.0.0 and GraphPad Prism 8.0.2. Statistical significance was set as $p < 0.05$.

# Results

## Description of study participants

The study recruited 938 children with a mean age (± standard deviation) of 6.4 (± 3.4) years. However, only 905 filter blots were found to be of good quality and thus used for this analysis.

**Table 1. Characteristics of study participants.**

| Variable | Keta (n = 272) | Hohoe (n = 327) | Krachi (n = 306) | Total (n = 905) | p-value |
|---|---|---|---|---|---|
| **Gender** | | | | | |
| Male | 134 (49.3) | 146 (44.6) | 147 (48.0) | 427 (47.2) | |
| Female | 138 (50.7) | 181 (55.4) | 159 (52.0) | 478 (52.8) | 0.50* |
| **Age Group (yrs)** | | | | | |
| Below 5 | 90 (33.1) | 165 (50.5) | 113 (36.9) | 368 (40.7) | |
| Above 5 | 182 (66.9) | 162 (49.5) | 193 (63.1) | 537 (59.3) | < 0.001* |
| **Ethnicity** | | | | | |
| Others | 1 (0.4) | 159 (48.6) | 127 (41.5) | 287 (31.7) | |
| Ewe | 264 (97.1) | 128 (39.1) | 25 (8.2) | 417 (46.1) | |
| Akan | 7 (2.6) | 7 (2.1) | 9 (2.9) | 23 (2.5) | |
| Guan | 0 (0.0) | 33 (10.1) | 145 (47.4) | 178 (19.7) | < 0.001* |
| **Bed net Usage** | | | | | |
| Yes | 258(94.9) | 282(86.2) | 281(91.8) | 821(90.7) | |
| No | 14(5.1) | 45(13.8) | 25(8.2) | 84(9.3) | 0.001* |
| **Hbg/dL(sd)** | 10.5 (1.3) | 11.2 (1.7) | 11.6 (1.6) | 11.2 (1.6) | < 0.001# |
| **Microscopy Parasite prevalence** | | | | | |
| Positive | 11(4.0) | 24(7.3) | 14(4.6) | 49(5.4) | |
| Negative | 261(96.0) | 303(92.7) | 292(95.4) | 856(94.6) | 0.15* |
| PCR Parasite prevalence | | | | | |
| Positive | 31(11.4) | 44(13.5) | 52(17.0) | 127(14.0) | |
| Negative | 241(88.6) | 283(86.5) | 254(83.0) | 778(86.0) | 0.14 |
| **Median parasite density/μl (iqr)** | 520 (220–7600) | 780 (350–3190) | 4660 (1360–45525) | 1280 (460–7680) | 0.020$ |

*Proportional differences between the three districts for gender, age groups, ethnicity, bed net usage, and parasite prevalence were determined using Chi-square test.

# the difference in mean haemoglobin (Hb) between districts was determined using ANOVA, and the Kruskal-Wallis test was used to calculate the differences in median parasite density

$. (%) Numbers in brackets are percentages. Iqr in the interquartile ranges. The "other" under the ethnicity is made up of minority groups who are not the predominant ethnic grouping in the areas studied. These include Hausa, Kotokoli, Zamrama, Mossi, Chamba, Yoruba, Fulani, Ga, Waala, Nawure, Kabre, Kokomba, and Tokosi

The three study sites contributed approximately the same number of participants, and the proportions of males and females were comparable. Children from Keta district had the lowest haemoglobin levels, compared to the other two districts. There were significant proportional differences among the districts in bed net usage. The prevalence of *P. falciparum* infection did not differ among the three sites, but the median parasite density was significantly higher at Krachi than the two other study sites. Parasite prevalence by PCR (already described in [23]) was higher than prevalence by microscopy. It did not however differ between the three sites (Table 1). None of the children was febrile, and all *P. falciparum* infections identified were asymptomatic.

## Seroprevalence and variation in IgG titres to malaria antigens

Generally, all children had detectable levels of IgG to the antigens tested. Antibody levels against all antigens tested increased with age in all the three districts. (Fig 2).

Seroprevalence was higher in Krachi for all the antigens tested, except for EBA140RIII-V, which had lower seroprevalence. The seroprevalence of MSPDBL$_{Leucine}$-, *Pf*Rh2b-, and anti-RON4-specific IgG were significantly higher in Krachi compared to Hohoe and Keta (Fig 3).

The highest antibody seroprevalence of 88.3% to MSPDBL$_{Leucine}$ and the lowest of 28.6% to EBA140RIII-V were observed in Krachi.

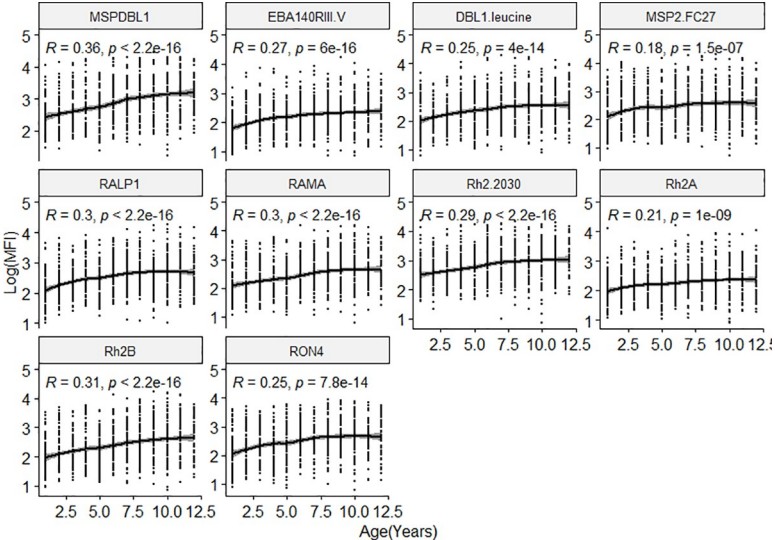

**Fig 2. Distribution of IgG levels (log-transformed MFI) as a function of age.** Overall differences of antibody levels to the ten antigens tested with age in the study population. The regression line shows the LOESS smoothed estimate of the log transformed MFI.

A comparison of antigen-specific antibody levels among the three districts showed that antibodies to MSPDBL1, MSPDDBL1$_{Leucine}$, MSP2$_{FC27}$, RAMA, $Pf$Rh2a and $Pf$Rh2b were all significantly higher in Krachi than in the other two districts. Keta had the lowest antibody levels except for $Pf$Rh2-$_{2030}$ (Fig 4).

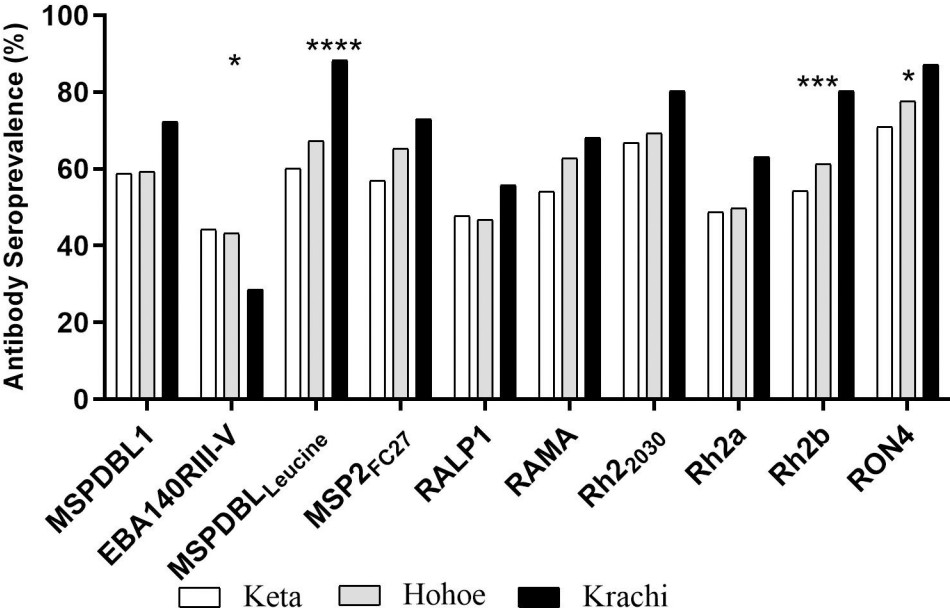

**Fig 3. Seroprevalence of antibodies found in the three districts.** Stars represent antigens with significantly higher proportions between the three districts using chi-square test. *p = <0.05, **p<0.01, ***p<0.001, ****p<0.0001. X-axis represent the antigens tested. Patterns represent study site. Seropositivity was defined as individuals whose mean total IgG levels to the antigens tested were higher than 2 standard deviations above the mean of malaria naïve individuals.

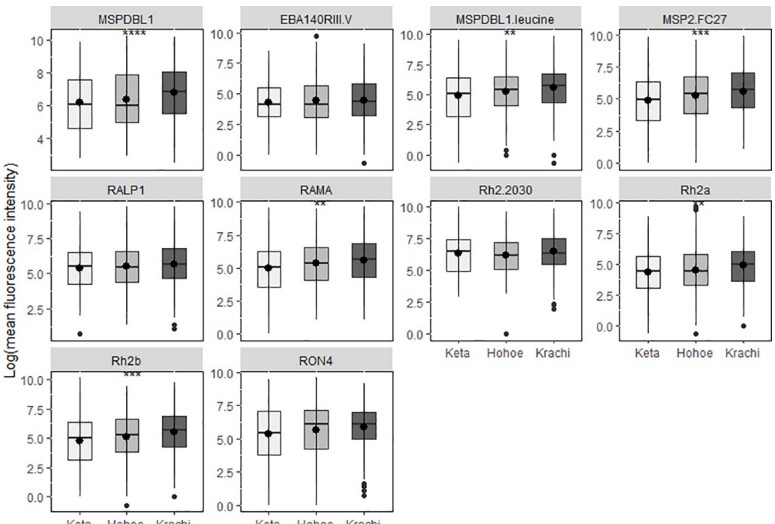

**Fig 4. Total IgG levels to multiple malaria antigens in the three districts.** Box and whisker plots with a round dot in the middle, the median IgG level of the group. Difference in antibody levels between the districts are shown for each antigen tested. The x-axis represents the districts (** p<0.01, *** p<0.001, ****p<0.0001).

## Antibody levels are associated with *P. falciparum* parasitaemia

To determine whether higher levels of antibodies were associated with parasite carriage, parasitaemia was determined by microscopy. Participants were then categorized, based on the presence or absence of parasites as a binary outcome variable, and antibody levels used as the predictor variable in a multivariable logistic regression analysis adjusting for district, age, ethnicity, and bed net usage. Overall, higher antibody levels to MSPDBL1 (odds ratio (OR) = 1.41, 95% confidence interval (CI) = 1.15 to 1.73), RAMA (OR = 1.32; CI = 1.06 to 1.63), *Pf*Rh2-$_{2030}$, (OR = 1.27; CI = 1.02 to 1.59), and *Pf*Rh2b (OR = 1.29; CI = 1.06 to 1.57) were associated with increased odds of *P. falciparum* parasite carriage. Also, a trend from low antibody levels in non-parasitaemic individuals to high antibody levels in individuals with submicroscopic parasitaemia was observed in individuals in Keta and Hohoe while no differences were observed in Krachi except antibodies to MSPDBL1 and *Pf*Rh2-$_{2030}$ (S1 Fig). Individuals with sub-microscopic parasitaemia are those with microscopy negative but are PCR positive. When antibody levels were compared between only microscopic and sub-microscopic groups, a trend of higher responses in the sub-microscopic group compared to the microscopic group was observed across the three sites. The responses were, however, not significant except MSP2$_{FC27}$ and *Pf*Rh2-$_{2030}$ which were significantly high in the sub-microscopic group in Krachi.

In a site-specific analysis, higher antibody responses to RALP-1 (OR = 1.93; CI = 1.14 to 3.26), *PfRh2a* (OR = 1.60; CI = 1.01 to 2.52), and *PfRh2b* (OR = 1.80; CI = 1.16 to 2.78), were associated with increased odds of carrying microscopic parasites in Keta. In Hohoe, the responses associated with parasite carriage were MSPDBL1 (OR = 1.56; CI = 1.18 to 2.06), MSPDBL$_{Leucine}$ (OR = 1.35; CI = 1.02 to 1.97), MSP2$_{FC27}$ (OR = 1.29; CI = 1.03 to 1.63), RAMA (OR = 1.56; CI = 1.17 to 2.08), *Pf*Rh2-$_{2030}$ (OR = 1.48; CI = 1.08 to 2.01), and *Pf*Rh2b (OR = 1.43; CI = 1.10 to 1.85). However, there was no association between parasite carriage and antibody levels in Krachi (p>0.05 for all antigens tested) (Fig 5).

In a linear model adjusting for age, ethnicity, bed net usage and haemoglobin levels to determine the relationship between district and parasitaemia, children in Krachi were found to have higher parasitaemia (β = 2.29; 95% CI = 0.56 to 4.02) than those in Hohoe. Also, sub-

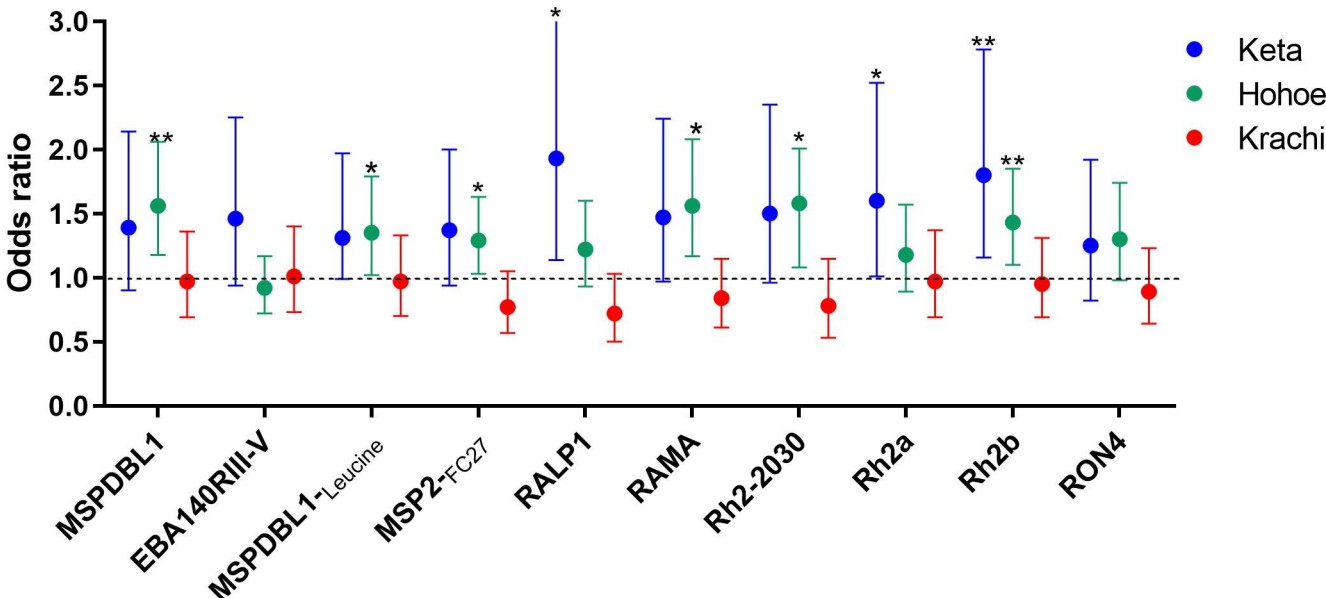

**Fig 5. Total IgG responses are associated with parasite carriage.** Multivariate logistic regression was used to predict parasite presence. Models were adjusted for age, ethnicity, bed net usage and district. Circles represent odds ratios and error bars are 95% confidence intervals. The black dotted line represents an OR of 1 which indicated no association with parasitaemia (*p = <0.05, **p<0.01, ***p<0.001, ****p<0.0001).

microscopic parasitaemia was associated with the high transmission district of Krachi (OR = 3.26; 95% CI = 1.39 to 7.64).

ROC curves were used to define a threshold MFI for predicting parasitaemia. The overall area under the curve (AUC) was lower than 66% for all antigens tested, however in a site-specific analysis, AUC for all the antigens in Keta was above 70% except RON4 67.6% (95% CI = 49.0 to 86.2) and MSPDBL1$_{Leucine}$ 68.2% (95% CI = 53.4 to 83.0), with *Pf*Rh2b having an AUC of 84.3 (95% CI = 76.3to 92.3). The AUC for all the antigens in the other districts were below 70% (Table 2). A threshold of *Pf*Rh2b levels at 639.5 MFI, was associated with a sensitivity of 88.9% (95% CI = 66.7–100) and a specificity of 79.4% (95% CI = 73.5–84.8) in predicting *P. falciparum* infection by microscopy with an AUC of 84.3%. The 95% CI were computed with 2,000 stratified bootstrap replicates.

**Table 2. Area under the curve (AUC) following receiver operating characteristics (ROC) analysis using antigens a predictor of parasitaemia.**

| Antigens | Keta AUC (95% CI) | Hohoe AUC (95% CI) | Krachi AUC (95% CI) |
|---|---|---|---|
| MSPDBL1 | 72.0 (56.8–87.1) | 69.7 (59.3–80.1) | 51.9 (41.6–62.2) |
| EBA140RIII-V | 71.8 (50.8–92.8) | 53.4 (42.5–64.2) | 52.5 (37.7–67.3) |
| MSPDBL$_{Leucine}$ | 68.2 (53.4–83.0) | 65.3 (54.1–76.5) | 52.4 (40.8–63.9) |
| MSP2$_{FC27}$ | 72.0 (62.6–81.3) | 64.9 (54.0–75.8) | 63.9 (51.9–75.9) |
| RALP1 | 76.7 (66.3–87.1) | 59.7 (47.7–71.6) | 60.8 (48.4–73.2) |
| RAMA | 76.7 (59.1–94.3) | 69.0 (56.7–79.3) | 57.0 (45.0–69.0) |
| *Pf*Rh2$_{2030}$ | 74.3 (58.0–90.0) | 66.3 (55.7–76.9) | 56.4 (44.7–68.7) |
| *Pf*Rh2a | 75.5 (62.2–89.3) | 60.4 (46.4–74.4) | 52.9 (40.1–65.7) |
| *Pf*Rh2b | 84.3 (76.3–92.3) | 67.6 (56.9–78.4) | 51.6 (40.4–62.8) |
| RON4 | 67.6 (49.0–86.2) | 62.0 (50.5–73.6) | 58.4 (46.0–70.7) |

The 95% CI for the area under the curve was calculated with 2000 stratified bootstrap replicates.

**Table 3. Factors associated with antibody breadth score.**

| Variable | β Coefficient | CI | p-value |
| --- | --- | --- | --- |
| **Age** | | | |
| Below 5 yrs | Ref | | |
| Above 5 yrs | 4.67 | 3.58 to 5.77 | **< 0.0001** |
| **Ethnicity** | | | |
| Akan | Ref | | |
| Ewe | -4.27 | -7.73 to -0.81 | **0.0158** |
| Guan | -3.10 | -6.69 to 0.48 | 0.0899 |
| others | -3.37 | -6.87 to 0.14 | 0.0599 |
| **Parasitaemia** | | | |
| Negative | Ref | | |
| Positive | 3.91 | 1.53 to 6.29 | **0.0013** |
| **PCR** | | | |
| Negative | Ref | | |
| Positive | 4.46 | 2.55 to 6.37 | **< 0.0001** |

Multiple linear regression was used to generate coefficient and CI.

## Association between breadth of antibody response and malaria transmission

Antibody responses to all the 10 antigens were scored depending on the quartiles for each individual and summed up to give the breadth score (number of antigens recognized). The breadth of antibody response in this study was between zero (0) and 30 with a breadth of 30 being the highest. Age, ethnicity and parasitaemia (by microscopy and PCR) factors were used in a linear model to study their relationship with breadth score. Parasitaemia was positively associated with high breadth score, while children above 5years of age had higher breadth score. Ethnicity (Ewe) was also negatively correlated with breadth score (Table 3).

To determine if transmission intensity was associated with antibody breadth score, the relationship between breadth score and malaria transmission (districts) was assessed in a linear model adjusting for age and ethnicity. Breadth score was positively associated with areas of higher malaria transmission (Krachi and Hohoe) compared to the low transmission area (Keta) (Table 4).

## Discussion

*P. falciparum* antigen-specific serology has been proposed as a tool to reduce the challenges of malaria transmission monitoring [4,8,32]. A key advantage of serology in the estimation of malaria transmission intensity is the ability to test large populations using dried spot samples with responses to multiple antigens using a multiplex assay approach [27,33]. To examine

**Table 4. Association between malaria transmission and antibody breadth score.**

| District | β Coefficient | CI | p-value |
| --- | --- | --- | --- |
| Keta | Ref | | |
| Hohoe | 4.18 | 2.56–5.79 | **< 0.0001** |
| Krachi | 4.53 | 2.56–6.50 | **< 0.0001** |

Multiple linear regression adjusting for age and ethnicity was used to generate coefficient and CI.

their suitability for this purpose, we determined antibody responses to multiple malaria antigens in three ecologically distinct districts of eastern Ghana with varying malaria transmission intensities.

The overall parasite prevalence (5.4%) was not significantly different among the three districts. The observed prevalence in Hohoe (7.3%) was lower than reported recently (16%) [34], which may be a result of ongoing malaria interventions in the area [35]. The observed prevalence (4.0%) in Keta was similar to that reported previously (3.7%) from a nearby site in southern Ghana with similar transmission pattern [36]. The similar prevalence in all the three study sites could be the due to the various malaria prevention interventions, which may have erased previous differences. The higher parasite densities of asymptomatic infections in Krachi than at the other sites may reflect better immunological control of disease-causing mechanisms (anti-disease immunity) in the high-endemicity setting [37].

Generally, participants in the study responded to all ten antigens tested and the antigens positively correlated with each other ($r^2 > 0.4$, $p < 0.001$ for all the antigens tested). Six (MSPDBL1, MSPDBL$_{Leucine}$, MSP2$_{FC27}$, RAMA, *Pf*Rh2a, and *Pf*Rh2b), (Table 2) elicited responses that were higher in Krachi compared to the other two districts. Variation in malaria transmission in the different ecological zones in Ghana has been reported [3]. The observed differences in the antibody levels to some of the antigens measured within the three districts reflected their pattern of malaria transmission, supporting the proposed use of serology to monitor transmission [4]. Antibody levels increased with age, reflecting the accumulation of IgG to an increasing number of different parasite antigens with time, thereby increasing antibody breadth [38,39]. However, levels of IgG specific for EBA140RIII-V, *Pf*Rh2-$_{2030}$, RALP-1, and RON4, did not differ among the three districts, indicating that they may be less suitable for monitoring transmission intensity. Since sampling was done in the dry season, the antibodies to these antigens may have waned since responses to some antigens have shorter half-life [40,41]. Also, these IgG responses much promptly react to the transmission change, and predict more recent transmission status. Seroprevalence of IgG to EBA140RIII-V was significantly lower in the high transmission area of Krachi, which is in contrast to earlier reports from Ghana [42]. However, it must be noted that differences in IgG quantification methods (bead-based array versus ELISA) may have accounted for the observed differences. Also, polymorphisms in EBA140III-V in parasite strains in the district may alter host receptor binding [43] and thus reduce their recognition in the district.

Infection with *P. falciparum* is generally thought to boost parasite-specific antibody responses by about 20% [44]. In the current study, parasitaemic (microscopic and sub-microscopic individuals) in both Hohoe and Keta districts had higher antibody levels compared to the non-parasitaemic individuals (S1 Fig) indicating that persisting parasitaemia does trigger continuous antibody production [45,46]. Also, sub-microscopic individuals with higher antibody levels may have a better control of parasitaemia reflecting in the low parasite levels. The lower antibody levels in the microscopic individuals compared to the sub-microscopic may indicate a low threshold not enough to control parasitaemia hence the high parasite levels diagnosed with microscopy. Since sub-microscopic parasitaemia is responsible for about 20% to 50% of human to mosquito infections [47], our finding that sub-microscopic parasitaemia is associated with Krachi which is a high transmission district corroborate earlier reports that sub-microscopic parasitaemia in asymptomatics could be a major factor in malaria transmission. It has also been suggested that polyclonal infections increase the breadth of antibody responses reflecting exposure and thus reduced risk of disease [48,49]. Thus, the high antibody responses in individuals from Krachi may be as a result of immune tolerance. These may indicate that a threshold of antibody level is needed to control parasitaemia and thus protection [50]. In a study conducted in Uganda, higher antibody levels against blood-stage antigens were

found to be protective against malaria symptoms once the subjects were parasitaemic [51]. Although the current study was a cross-sectional study, with the limitation that we could not follow the participants to know the exact point at which infection occurred, we believe higher antibody levels in the parasitaemic children could reflect higher transmission. The similar antibody levels between parasitaemic and non-parasitaemic children in Krachi may be as a result of the high transmission reflecting in high parasite density which is associated with high sub-microscopic parasitaemia and polyclonality often expressed as multiplicity of infection (MOI) [52,53]. Antibody levels expressed by both parasitaemic and non-parasitaemic but high MOI individuals may be almost the same since high sub-microscopic parasitaemia may also induce higher responses. Sub-microscopic parasitaemia is about 50% higher than microscopy [54]. It could also be as a result of maintenance of antibodies from the rainy season. We could not collect data for entomological inoculation rate (EIR) during sampling, which is another limitation but the low to high antibody levels from Keta to Krachi, respectively, may indicate transmission differences which reflects antibody levels in the three districts.

This study also found site-specific antibody responses to associate with *P. falciparum* carriage (Fig 5). Different responses were associated with *P. falciparum* carriage in both Keta and Hohoe, although levels of *Pf*Rh2b-specific IgG were associated with parasite carriage in both districts. Keta and Hohoe districts both have lower malaria transmission than Krachi, and the association of *Pf*Rh2b with parasitaemia indicates a possible use of that antigen to monitor malaria transmission and pattern in low transmission setting. The different antigens predicting parasite carriage in the two districts with different transmission intensities reflect possible differences in responses of individuals within each transmission zone. We used ROC analysis to confirm if indeed *Pf*Rh2b and any other antigen could be used to monitor transmission and pattern in a low transmission setting. We found that IgG responses to *Pf*Rh2b at a level of 639.5 MFI in low transmission settings are likely to predict *P. falciparum* carriage by microscopy further confirming the possible use of *Pf*Rh2b to monitor transmission and pattern.

Several reports have linked breadth of antibody responses to malaria immunity and that individuals with higher antibody breadth have better protection from malaria [18,55]. A study in three highly endemic districts in Ghana identified age and endemicity as predictors of antibody breadth [38]. Thus, the relationship between age, ethnicity and parasitaemia was explored. Our observation of a low to high trend of antibody breadth from keta to Krachi suggests that breadth may be linked to transmission intensity and pattern. The low antibody breadth score in children living in Keta could be linked to the low transmission intensity resulting in low antibody responses in the district.

Though antibody levels may be higher in the rainy season, the data presented here demonstrate variations in antibody response to multiple malarial antigens in children across different ecological regions with varying transmission intensities, confirming previous reports of serology as an alternative malaria transmission monitoring tool [4,8,9].

In conclusion, the study highlights the potential of antibodies against *Pf*Rh2b as a useful marker for predicting malaria transmission intensity and pattern in low malaria transmission setting. Antibody levels and seroprevalence reflects changes in transmission between the three ecological zones. Low breadth score was found to be associated with low malaria transmission. The study findings have implications for malaria control interventions. Also, the design and testing of vaccines must take into consideration the heterogeneity of immune responses in the different ecological zones.

## Supporting information

**S1 Fig. Total IgG responses between parasitaemic and non-parasitaemic individuals.** Box and whisker plots with a round dot in the middle, the median IgG level of the group.

Differences in antibody levels between non-parasitaemic, microscopic, and sub-microscopic individuals for each antigen are shown for each district. The x-axis represents the districts, and the y-axis represent the log-transformed antibody levels (** p<0.01, ***p<0.001, ****p<0.0001).
(TIF)

## Acknowledgments

The Authors are grateful to the parents and guardians of the children involved in the study. We also thank the District and Municipal Directors of Health in Keta, Hohoe, and Krachi West Districts for supporting the team during sampling. We are also thankful to Mr Jones Amo Amponsah and Alex Danso-Coffie of the Immunology Department, NMIMR for his technical assistance.

## Author Contributions

**Conceptualization:** Eric Kyei-Baafour, Margaret Kweku, Lars Hviid, Michael Fokuo Ofori.

**Data curation:** Eric Kyei-Baafour, Mavis Oppong, Abena Fremaah Frempong, Belinda Aculley, Fareed K. N. Arthur, Margaret Kweku, Bright Adu, Michael Fokuo Ofori.

**Formal analysis:** Eric Kyei-Baafour, Mavis Oppong, Kwadwo Asamoah Kusi, Abena Fremaah Frempong, Belinda Aculley, Bright Adu, Lars Hviid, Michael Fokuo Ofori.

**Funding acquisition:** Bright Adu, Lars Hviid, Michael Fokuo Ofori.

**Project administration:** Michael Fokuo Ofori.

**Resources:** Regis Wendpayangde Tiendrebeogo, Susheel K. Singh, Michael Theisen, Lars Hviid, Michael Fokuo Ofori.

**Supervision:** Kwadwo Asamoah Kusi, Fareed K. N. Arthur, Michael Theisen, Margaret Kweku, Bright Adu, Michael Fokuo Ofori.

**Writing – original draft:** Eric Kyei-Baafour, Mavis Oppong, Kwadwo Asamoah Kusi, Abena Fremaah Frempong, Belinda Aculley, Fareed K. N. Arthur, Regis Wendpayangde Tiendrebeogo, Susheel K. Singh, Michael Theisen, Margaret Kweku, Bright Adu, Lars Hviid, Michael Fokuo Ofori.

**Writing – review & editing:** Eric Kyei-Baafour, Mavis Oppong, Kwadwo Asamoah Kusi, Abena Fremaah Frempong, Belinda Aculley, Fareed K. N. Arthur, Regis Wendpayangde Tiendrebeogo, Susheel K. Singh, Michael Theisen, Margaret Kweku, Bright Adu, Lars Hviid, Michael Fokuo Ofori.

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
