## [Decision Letter · Decision Letter 0]

8 Jan 2021

PONE-D-20-38844

Suitability of IgG responses to multiple Plasmodium falciparum antigens as markers of transmission intensity and pattern

PLOS ONE

Dear Dr. Ofori,

Thank you for submitting your manuscript to PLOS ONE. After careful consideration, we feel that it has merit but does not fully meet PLOS ONE’s publication criteria as it currently stands. Therefore, we invite you to submit a revised version of the manuscript that addresses the points raised during the review process.

Please respond to all of the comments from the Reviewers and revise the manuscript.

We look forward to receiving your revised manuscript.

Kind regards,

Takafumi Tsuboi

Academic Editor

PLOS ONE

Journal Requirements:

2. In your Methods section, please provide additional information about the participant recruitment method and the demographic details of your participants. Please ensure you have provided sufficient details to replicate the analyses such as:

a) a description of any inclusion/exclusion criteria that were applied to participant recruitment,

b) a statement as to whether your sample can be considered representative of a larger population, and

c) a description of how participants were recruited.

3.  We note that Figure 1 in your submission contain map images which may be copyrighted. All PLOS content is published under the Creative Commons Attribution License (CC BY 4.0), which means that the manuscript, images, and Supporting Information files will be freely available online, and any third party is permitted to access, download, copy, distribute, and use these materials in any way, even commercially, with proper attribution. For these reasons, we cannot publish previously copyrighted maps or satellite images created using proprietary data, such as Google software (Google Maps, Street View, and Earth). For more information, see our copyright guidelines: http://journals.plos.org/plosone/s/licenses-and-copyright.

3.1.    You may seek permission from the original copyright holder of Figure 1 to publish the content specifically under the CC BY 4.0 license. 

3.2.    If you are unable to obtain permission from the original copyright holder to publish these figures under the CC BY 4.0 license or if the copyright holder’s requirements are incompatible with the CC BY 4.0 license, please either i) remove the figure or ii) supply a replacement figure that complies with the CC BY 4.0 license. Please check copyright information on all replacement figures and update the figure caption with source information. If applicable, please specify in the figure caption text when a figure is similar but not identical to the original image and is therefore for illustrative purposes only.

Reviewers' comments:

Reviewer's Responses to Questions

**Comments to the Author**

1. Is the manuscript technically sound, and do the data support the conclusions?

Reviewer #1: Yes

Reviewer #2: Partly

2. Has the statistical analysis been performed appropriately and rigorously? 

Reviewer #1: Yes

Reviewer #2: Yes

3. Have the authors made all data underlying the findings in their manuscript fully available?

Reviewer #1: Yes

Reviewer #2: Yes

4. Is the manuscript presented in an intelligible fashion and written in standard English?

Reviewer #1: Yes

Reviewer #2: Yes

5. Review Comments to the Author

Reviewer #1: This study “Suitability of IgG responses to multiple Plasmodium falciparum antigens as markers of transmission intensity and pattern” by Kyei-Baafour et al sought to assess the suitability of P. falciparum parasite specific antibodies as markers of transmission intensity and pattern. They used a multiplex assay to evaluate Antibodies (Abs) responses to 10 antigens and examined their relationships with age, parasitemia, and the study sites. They concluded that PfRh2b has potential as a marker of malaria transmission intensity and pattern.

The manuscript is well written with few errors. Although the conclusions are largely valid they are not novel. Nevertheless, there are several issues that authors need to address in order to improve the findings.

Introduction

1. The authors should consider making line 77-87 as a stand-alone paragraph to explain the what constitute a good Abs marker of transmission intensity and pattern.

2. Lines 57-60 would best be moved and combined with line 88 to clearly show that existing gap.

3. Line 48 needs to be corrected. ..“In Africa, 96% of malaria cases are due to Plasmodium falciparum of the cases in 2019”.

Methods

1. In general, the authors should make it clear – at least briefly - to the readers what was done without having to do a lot of cross-referencing with previous study.

2. Where were the DBS stored? A-20°C or at 4oC as previously reported?

3. This study is well meaning, however, between 6 and 14 years had passed since the transmission intensities (as referenced 5, 6 &7) were documented and the time of samples collection. In these years, a lots of malaria control interventions were conducted in the same regions mainly driven by the EIR data. This would translate that the EIR were very different or even opposite. The conclusions would be more informative if accompanied by the corresponding EIR taken during sample collection.

4. The study lacks a clear rationale of why the 10 proteins were selected over other parasites >5000 proteins and why other more common protein families such as VSAs were not considered.

5. And, how were the 10 proteins were produced, and which parasite variant(s) were they based on? This is important for discussion study.

6. Which QC measures were put in place to make sure same amount of serum was collected and eluted from each DBS between individuals and across the three different site?

7. How did the authors adjust or normalize the amount of hemoglobin co-eluted with serum since high levels of hemoglobin, as expected in some samples, may disrupt antibody binding or clog the probe during Luminex resulting in lower signals in those samples? Lack of this normalization introduces uncertainty into the data.

8. What was the source of the malaria-naïve control samples ? How was the seropositivity cut-off determined? Was it antigen- specific or study specific? This need to be clarified.

Results

1. Table 1. the presentation of P values needs to be standardized i.e. < 0.0001 or < 1e-04.

2. The tense in line 187 need to be in line with other results.

3. Fig 2. Standardize the y-axis i.e. 10 or 10.0.

4. Fig 2. I think it would be more informative to show a matrix of correlation scatter plots each with a regression line between age and MFI since the two are continues variables. Ages 1-5 and 5-15 are quite diverse and a lot may be hidden by the current presentation.

190. In this population, we would expect that a great percentage of the older children would have submicroscopic infection and all P. falciparum infections identified were asymptomatic. The authors seems not to have considered the effect of this submicroscopic infections on antibody responses. In addition, they should further evaluate the effect of Abs on increasing parasite density to access the ability of individuals to control the parasitemia.

Discussion

1. Since the main objective of this study was to identify potential markets of malaria transmission, submicroscopic infections should be considered and discussed throughout the study. This is expected to exist in a region of high malaria transmission, and is considered to be a major contributor of gametocytes necessary for transmission. Several researchers have already reported this.

https://malariajournal.biomedcentral.com/articles/10.1186/s12936-016-1482-4

https://malariajournal.biomedcentral.com/articles/10.1186/s12936-018-2479-y

https://www.nature.com/articles/s41598-019-53386-w

2. Line: 359 – 366: Again, the authors would like to associate the Abs data with EIR collected many years earlier. This may be very misleading and needs to be further explained or removed.

Reviewer #2: [Remarks to the Author]

This manuscript described on the results of seroepidemiological survey targeting for multiple malaria antigens in the 1-12 years old participants in the Volta region of Ghana. By comparing the antibody level, seroprevalence, and the breadth of antibody response to 10 malaria blood stage antigens with age, endemicity, parasitic status, the authors characterized differences in malaria-specific antibody responses. Furthermore receiver operating characteristics analysis was performed using antigens a predictor of parasitemia and proposed that PfRh2b has potential as a malaria of malaria transmission intensity. The serological analysis has been performed in an appropriate manner, however I think the authors can revisit the analysis purpose and discussion once again as indicating in the following comments.

[Major comments]

1. One of the aim of this study is to test whether the reaction to these antibody can guide us to predict the difference of malaria transmission, and the authors tried to investigate it by focusing on three different areas in the Volta region of Ghana. The authors claimed that these three categorized in three different ecological zones having different malaria transmission intensity. However the provided results of malaria prevalence showed pretty much similar level of it in the all areas. Although the authors discussed that this might be due to the various malaria prevention interventions and that the difference in the parasite densities might reflect the difference of the previous transmission, it still supports the transmission intensity has changed recently. If we want to discuss the usefulness of serology as a prediction of the transmission, I think we cannot ignore this fact. In other way around, some antibodies which the authors did not find the difference between the area may much sensitively reflect the change of the transmission. Furthermore the antibodies have a variation in the period of remaining in the blood stream, thus (a) the clarification of transmission dynamics in a time course, (b) the clarification of focus (which time point the authors focus on and what kind of prediction model the authors have in mind) and (c) discussion taking into account for the antibody lasting period with transmission course may need to make the discussion sense.

2. The authors concluded PfRh2b as a potential marker of malaria transmission intensity and pattern based on the association of it with parasite carriage. However I think this is a discussion confusing the prediction of individuals and population. The analysis performed here is to see that the antibody reaction can distinguish malaria parasite carrier from no-carrier, and even though the high transmission intensity is correlated to the accumulation of parasite carriers, these are different parameter, thus we cannot conclude in that way. Furthermore the parasitic status is a single point evaluation, thus linking the serological value with this is logically unexplainable even though there are correlation.

[Minor comments]

l.56 Bringing parasite prevalence prior to EIR may be more understandable as a general discussion. Or basically, to predict the prevalence, we try to use EIR or serology.

l.81 The reason why these antigens were selected are not clear, especially in the light of research objectives. This helps us to understand the objectives of study.

l.96 How were “random” selection performed?

Fig.1 The legend should also mention sub-region.

l.123 I think it is “1000x magnification”. Did you exam with thick smear or thin smear?

l.151 What is the definition of the malaria naïve individuals?

l.165 How many naïve samples were included to get the mean?

l.177 Typo of the end of the parenthesis

Table.1 What is the number in the bracket of Hb?

Figure.3 MSP1DBL-Leucine> MSPDBL-Leucine

l.264 What is the purpose of performing this linear model analysis?

l.317 I believe the advantage of serology is that it can predict and study the trend and/or history of the infection in the target population from a single point survey. Estimating the prevalence of that timing by testing large population with DBS can be done with normal prevalence survey.

l.339 As mentioned major comment #1, probably we could rather say that these IgG much promptly react to the transmission change, and predict more recent transmission status.

l.339 duplicated “however”

l.355 Why this hypothesis only applied for Krachi? These phenomenon may occur in other area as well, thus this explanation may not be appropriate for.

l.371 The key message from this and the following paragraph was not clear to me.

6. PLOS authors have the option to publish the peer review history of their article (what does this mean?). If published, this will include your full peer review and any attached files.

Reviewer #1: No

Reviewer #2: **Yes: **Wataru Kagaya

---

## [Author Response · Author response to Decision Letter 0]

18 Feb 2021

Title of paper: 

Suitability of IgG responses to multiple Plasmodium falciparum antigens as markers of transmission intensity and pattern

Kyei-Baafour et al.

Dear Editor,

Thank you for your email on 8th January 2021, in which you informed us that our manuscript has been peer reviewed and that it may be considered for publication after we have revised the manuscript as suggested by the reviewers (Manuscript ID 295277), title above. We thank the Editor and reviewers for their critical review and additional comments which we know when addressed will improve the manuscript. 

We hereby submit for your consideration a revised version, in which we have carefully considered all the comments and suggestions made by the reviewers.

Please find below our point-by-point response to editorial corrections and reviewers’ comments, and a tracked changes version of the revised manuscript. 

Thank you.

Yours Sincerely,

Michael F. Ofori

1. Please ensure that your manuscript meets PLOS ONE’s style requirements, including those for file naming. 

Response: The manuscript has been reformatted to meet the PLOS ONE’s style including the

 naming of files

2. In your Methods section, please provide additional information about the participant recruitment method and the demographic details of your participants. Please ensure you have provided sufficient details to replicate the analyses such as:

a) a description of any inclusion/exclusion criteria that were applied to participant recruitment,

b) a statement as to whether your sample can be considered representative of a larger population, and

c) a description of how participants were recruited.

Response: Statements have been added from Line 100-108 describing how the three districts were selected to reflect the volta region, and how the participants were recruited with the inclusion and exclusion criteria.

3. We note that Figure 1 in your submission contain map images which may be copyrighted. All PLOS content is published under the Creative Commons Attribution License (CC BY 4.0), which means that the manuscript, images, and Supporting Information files will be freely available online, and any third party is permitted to access, download, copy, distribute, and use these materials in any way, even commercially, with proper attribution. For these reasons, we cannot publish previously copyrighted maps or satellite images created using proprietary data, such as Google software (Google Maps, Street View, and Earth). For more information, see our copyright guidelines: http://journals.plos.org/plosone/s/licenses-and-copyright.

3.1. You may seek permission from the original copyright holder of Figure 1 to publish the content specifically under the CC BY 4.0 license. 

3.2. If you are unable to obtain permission from the original copyright holder to publish these figures under the CC BY 4.0 license or if the copyright holder’s requirements are incompatible with the CC BY 4.0 license, please either i) remove the figure or ii) supply a replacement figure that complies with the CC BY 4.0 license. Please check copyright information on all replacement figures and update the figure caption with source information. If applicable, please specify in the figure caption text when a figure is similar but not identical to the original image and is therefore for illustrative purposes only.

Response: The map image was design in-house by Jessica Asante using QGIS version 3.4.7 and a line has been inserted in Line 120-121 crediting her.

Reviewer #1: This study “Suitability of IgG responses to multiple Plasmodium falciparum antigens as markers of transmission intensity and pattern” by Kyei-Baafour et al sought to assess the suitability of P. falciparum parasite specific antibodies as markers of transmission intensity and pattern. They used a multiplex assay to evaluate Antibodies (Abs) responses to 10 antigens and examined their relationships with age, parasitemia, and the study sites. They concluded that PfRh2b has potential as a marker of malaria transmission intensity and pattern.

The manuscript is well written with few errors. Although the conclusions are largely valid they are not novel. Nevertheless, there are several issues that authors need to address in order to improve the findings.

Introduction

1. The authors should consider making line 77-87 as a stand-alone paragraph to explain the what constitute a good Abs marker of transmission intensity and pattern.

Response: A paragraph has been created as suggested by reviewer on Line 75-85

2. Lines 57-60 would best be moved and combined with line 88 to clearly show that existing gap.

Response: The paragraph (Lines 57-60) has been moved and has now been combined with the last paragraph (Lines 86 -93)

3. Line 48 needs to be corrected. ..“In Africa, 96% of malaria cases are due to Plasmodium falciparum of the cases in 2019”.

Response: Line 49 has been corrected to read “In Africa, 96% of malaria cases were due to Plasmodium falciparum in 2019”

Methods

1. In general, the authors should make it clear – at least briefly - to the readers what was done without having to do a lot of cross-referencing with previous study.

Response: the sampling method employed and inclusion/exclusion criteria have been inserted on lines 100-111

2. Where were the DBS stored? A-20°C or at 4oC as previously reported?

Response: Where the DBS were stored has been indicated (line 129)

3. This study is well meaning, however, between 6 and 14 years had passed since the transmission intensities (as referenced 5, 6 &7) were documented and the time of samples collection. In these years, a lots of malaria control interventions were conducted in the same regions mainly driven by the EIR data. This would translate that the EIR were very different or even opposite. The conclusions would be more informative if accompanied by the corresponding EIR taken during sample collection.

Response: A limitation was the lack of EIR data in the current study. However, the antibody levels to almost all the antigens tested clearly shows a pattern of low responses from Keta to high responses in Krachi which may reflect differences in transmission intensity and a statement has been inserted in Line 378-381 to reflect this limitation.

4. The study lacks a clear rationale of why the 10 proteins were selected over other parasites >5000 proteins and why other more common protein families such as VSAs were not considered.

Response: They were selected to test our hypothesis about whether the position of a protein on the merozoite could help in determining its usefulness as a transmission monitoring marker. Some of these antigens have also been associated with protection by Ghanaian and Indian cohorts. A statement has been inserted that describes the rationale for selecting these proteins (Line 152 – 153).

5. And, how were the 10 proteins were produced, and which parasite variant(s) were they based on? This is important for discussion study.

Response: A statement has been inserted explaining how all the proteins were produced and the variants from which they were based. Line 150-152.

6. Which QC measures were put in place to make sure same amount of serum was collected and eluted from each DBS between individuals and across the three different site?

Response: For each DBS, the same 2.5 mm diameter of cuts were made (on line 143) and the same volume of elution buffer (150µl) was added to the cut DBS (on line 144). This gave a dilution of about 1:100 and this dilution was taken into consideration in determining the dilution for the final Luminex assay. A line has been inserted (Line 168) to reflect the final dilution in the Luminex assay. 

7. How did the authors adjust or normalize the amount of hemoglobin co-eluted with serum since high levels of hemoglobin, as expected in some samples, may disrupt antibody binding or clog the probe during Luminex resulting in lower signals in those samples? Lack of this normalization introduces uncertainty into the data.

Response: data was normalized for inter-plate and day-to-day variation by dividing the test sample on each plate by the mean positive control of the assay plate. This was multiplied by the total mean positive control for all the plates to obtain the normalized value for the sample using the formula: 

(Sample/Plate Positive control) x Mean positive control for all plates

A statement has been added that explains how the normalization was done (Line 181-184) 

8. What was the source of the malaria-naïve control samples ? How was the seropositivity cut-off determined? Was it antigen- specific or study specific? This need to be clarified.

Response: The source of the malaria naïve control sera has been provided (Line 166 -167). Seropositivity was antigen-specific and was determined before all the analysis. 

A statement has been inserted that explains how the seropositivity cut-off was determined and the fact that the seroprevalence was antigen-specific (Lines 185-187).

Results

1. Table 1. the presentation of P values needs to be standardized i.e. < 0.0001 or < 1e-04.

Response: P-values have been standardized in Table 1 

2. The tense in line 187 need to be in line with other results.

Response: The tense has been corrected to reflect the suggestion (Line 208).

3. Fig 2. Standardize the y-axis i.e. 10 or 10.0.

Response: the y-axis of Figure 2 has been log transformed and standardized 

4. Fig 2. I think it would be more informative to show a matrix of correlation scatter plots each with a regression line between age and MFI since the two are continues variables. Ages 1-5 and 5-15 are quite diverse and a lot may be hidden by the current presentation.

Response: Figure 2 has been replaced with a correlation scatter plot for each antigen tested with a regression line showing the coefficient and p-value as suggested by Reviewer 

190. In this population, we would expect that a great percentage of the older children would have submicroscopic infection and all P. falciparum infections identified were asymptomatic. The authors seems not to have considered the effect of this submicroscopic infections on antibody responses. In addition, they should further evaluate the effect of Abs on increasing parasite density to access the ability of individuals to control the parasitemia.

Response: A statement has been inserted that considers the effect of submicroscopic infection on antibody responses. (Line 270-274).

Discussion

1. Since the main objective of this study was to identify potential markets of malaria transmission, submicroscopic infections should be considered and discussed throughout the study. This is expected to exist in a region of high malaria transmission, and is considered to be a major contributor of gametocytes necessary for transmission. Several researchers have already reported this.

https://malariajournal.biomedcentral.com/articles/10.1186/s12936-016-1482-4

https://malariajournal.biomedcentral.com/articles/10.1186/s12936-018-2479-y

https://www.nature.com/articles/s41598-019-53386-w

Response: Analysis of the parasitaemic individuals taking into consideration sub-microscopic has been included in the manuscript. Tables 1 and 3 have been updated with PCR parasitaemia, supplementary figure 1 has also been updated to include effects on sub-microscopic parasitaemia on antibody levels across the three districts.

The results section has been updated (Line 270-274) likewise the discussion (lines 387 and 390 - 394) also updated. 

2. Line: 359 – 366: Again, the authors would like to associate the Abs data with EIR collected many years earlier. This may be very misleading and needs to be further explained or removed.

Response: Response: A statement of limitation has been inserted that explains that data was not collected on entomological inoculation rates in this study (Lines 405-408)

Reviewer #2: [Remarks to the Author]

This manuscript described on the results of seroepidemiological survey targeting for multiple malaria antigens in the 1-12 years old participants in the Volta region of Ghana. By comparing the antibody level, seroprevalence, and the breadth of antibody response to 10 malaria blood stage antigens with age, endemicity, parasitic status, the authors characterized differences in malaria-specific antibody responses. Furthermore receiver operating characteristics analysis was performed using antigens a predictor of parasitemia and proposed that PfRh2b has potential as a malaria of malaria transmission intensity. The serological analysis has been performed in an appropriate manner, however I think the authors can revisit the analysis purpose and discussion once again as indicating in the following comments.

[Major comments]

1. One of the aim of this study is to test whether the reaction to these antibody can guide us to predict the difference of malaria transmission, and the authors tried to investigate it by focusing on three different areas in the Volta region of Ghana. The authors claimed that these three categorized in three different ecological zones having different malaria transmission intensity. However the provided results of malaria prevalence showed pretty much similar level of it in the all areas. Although the authors discussed that this might be due to the various malaria prevention interventions and that the difference in the parasite densities might reflect the difference of the previous transmission, it still supports the transmission intensity has changed recently. If we want to discuss the usefulness of serology as a prediction of the transmission, I think we cannot ignore this fact. In other way around, some antibodies which the authors did not find the difference between the area may much sensitively reflect the change of the transmission. Furthermore the antibodies have a variation in the period of remaining in the blood stream, thus (a) the clarification of transmission dynamics in a time course, (b) the clarification of focus (which time point the authors focus on and what kind of prediction model the authors have in mind) and (c) discussion taking into account for the antibody lasting period with transmission course may need to make the discussion sense.

Response A: The rainfall pattern that reflects transmission dynamics in the study areas have been described fully under the methods section. (Lines 109-116)

Response: B, the study was conducted in December and that constituted the time point for the prediction (Line 97). 

Response C: A statement has been made to reflect the suggestion made by the reviewer ( 378-380)

2. The authors concluded PfRh2b as a potential marker of malaria transmission intensity and pattern based on the association of it with parasite carriage. However, I think this is a discussion confusing the prediction of individuals and population. The analysis performed here is to see that the antibody reaction can distinguish malaria parasite carrier from no-carrier, and even though the high transmission intensity is correlated to the accumulation of parasite carriers, these are different parameter, thus we cannot conclude in that way. Furthermore, the parasitic status is a single point evaluation, thus linking the serological value with this is logically unexplainable even though there are correlation.

Response: the analysis was to distinguish which antibody can distinguish or predict Plasmodium carrier from a non-carrier by microscopy. There were a number of them that came out (Fig 5), however a more robust analysis (Roc analysis) was done to confirm the findings in figure 5 and only Rh2b came out clearly to distinguish the two. This was stated clearly in the results section and discussed as well. We believe the analysis done suited our research questions.

[Minor comments]

l.56 Bringing parasite prevalence prior to EIR may be more understandable as a general discussion. Or basically, to predict the prevalence, we try to use EIR or serology.

Response: The statement has been modified as suggested by reviewer to read (Line 56-58).

l.81 The reason why these antigens were selected are not clear, especially in the light of research objectives. This helps us to understand the objectives of study.

Response: They were selected to test our hypothesis about whether the position of a protein on the merozoite could help in determining its usefulness as a transmission monitoring marker. Some of these antigens have also been associated with protection by Ghanaian and Indian cohorts. A statement has been inserted that describes the rationale for selecting these proteins (Line 152 – 153).

l.96 How were “random” selection performed?

Response: The sampling and selection method employed has been explained (Lines 103-111

Fig.1 The legend should also mention sub-region.

Response: The legend of Figure 1 has been updated to include the sub-region as suggested 

 (Line 117) 

l.123 I think it is “1000x magnification”. Did you exam with thick smear or thin smear?

Response: It was 100X magnification (Line 134) and the statement on line 135 has also been modified that states the use of the thick smear in the estimation.

l.151 What is the definition of the malaria naïve individuals?

Response: Astaement has been inserted that describe what exactly naïve individuals mean (Line 166-168

l.165 How many naïve samples were included to get the mean?

Response: The total number of naïve samples has been specified (Line 185)

l.177 Typo of the end of the parenthesis

Response: Typo has been corrected (Line197)

Table.1 What is the number in the bracket of Hb?

Response: the number in the bracket of Hb is the standard deviation and Table 1 has been updated to reflect that.

Figure.3 MSP1DBL-Leucine> MSPDBL-Leucine

Response: Table 3 has been updated to reflect the actual name of the protein MSPDBL-Leucine

l.264 What is the purpose of performing this linear model analysis?

Response: The Linear model analysis was performed to determine the correlation between parasitaemia and the districts

l.317 I believe the advantage of serology is that it can predict and study the trend and/or history of the infection in the target population from a single point survey. Estimating the prevalence of that timing by testing large population with DBS can be done with normal prevalence survey.

Response: One advantage of serology in studying malaria transmission pattern using DBS is that some antibodies are short lived therefore any antibody signal detected as a result of recent exposure/infection and may not be affected by transmission. Normal prevalence survey using microscopy may not be sensitive in low transmission setting and that will also be labor intensive.

l.339 As mentioned major comment #1, probably we could rather say that these IgG much promptly react to the transmission change, and predict more recent transmission status.

l.339 duplicated “however”

Response: duplicated word has been removed

l.355 Why this hypothesis only applied for Krachi? These phenomenon may occur in other area as well, thus this explanation may not be appropriate for.

Response: The paragraph has been rephrased to address the concern raised by the reviewer (Line 400-408).

l.371 The key message from this and the following paragraph was not clear to me.

Response: The paragraph has been updated with a statement to make it clearer as suggested by the reviewer: (Line 424-426)

---

## [Decision Letter · Decision Letter 1]

15 Mar 2021

PONE-D-20-38844R1

Suitability of IgG responses to multiple Plasmodium falciparum antigens as markers of transmission intensity and pattern

PLOS ONE

Dear Dr. Ofori,

Thank you for submitting your manuscript to PLOS ONE. After careful consideration, we feel that it has merit but does not fully meet PLOS ONE’s publication criteria as it currently stands. Therefore, we invite you to submit a revised version of the manuscript that addresses the points raised during the review process.

Please take the minor comments from the Review 1 into consideration on the final revision.

We look forward to receiving your revised manuscript.

Kind regards,

Takafumi Tsuboi

Academic Editor

PLOS ONE

Journal Requirements:

Reviewers' comments:

Reviewer's Responses to Questions

**Comments to the Author**

1. If the authors have adequately addressed your comments raised in a previous round of review and you feel that this manuscript is now acceptable for publication, you may indicate that here to bypass the “Comments to the Author” section, enter your conflict of interest statement in the “Confidential to Editor” section, and submit your "Accept" recommendation.

Reviewer #1: (No Response)

Reviewer #2: All comments have been addressed

2. Is the manuscript technically sound, and do the data support the conclusions?

Reviewer #1: Yes

Reviewer #2: (No Response)

3. Has the statistical analysis been performed appropriately and rigorously? 

Reviewer #1: Yes

Reviewer #2: (No Response)

4. Have the authors made all data underlying the findings in their manuscript fully available?

Reviewer #1: Yes

Reviewer #2: (No Response)

5. Is the manuscript presented in an intelligible fashion and written in standard English?

Reviewer #1: Yes

Reviewer #2: (No Response)

6. Review Comments to the Author

Reviewer #1: Review.

The authors adequately responded to all comments. Inclusion of the sub-microscopic data clearly improves the impact of this manuscript but needs to be well discussed.

Just some minor comments

Line 270: 274: Was there any antibody difference between microscopic and submicroscopic groups? Since this data is available it’s should be mentioned as well as discussed for Keta and Hohoe vis a vis Krachi. And how does this affect or influence transmission?, and relate with the authors conclusion that, “These indicate that a threshold of antibody level is needed to control parasitaemia and thus protection”.

Line 309: MSPDBL_Leucine was not consistently revised correctly. Line 302 vs line 285 reads MSPDBL1_Leucine. Generally, all antigen names should be checked again.

Reviewer #2: (No Response)

7. PLOS authors have the option to publish the peer review history of their article (what does this mean?). If published, this will include your full peer review and any attached files.

Reviewer #1: **Yes: **Bernard N. Kanoi

Reviewer #2: No

---

## [Author Response · Author response to Decision Letter 1]

24 Mar 2021

Title of paper: PONE-D-20-38844R1

Suitability of IgG responses to multiple Plasmodium falciparum antigens as markers of transmission intensity and pattern

Kyei-Baafour et al.

Dear Editor,

Thank you for your email on 15 March 2021, in which you informed us that our manuscript after careful consideration, you feel that it has merit but does not fully meet PLOS ONE’s publication criteria as it currently stands. Therefore, you invited us to submit a revised version of the manuscript that addresses the points raised during the review process. (Manuscript ID PONE-D-20-38844R1)), title above.

 We thank the Editor and reviewers for their critical review and additional comments which we know when addressed will improve the manuscript. 

We hereby submit for your consideration a revised version, in which we have carefully considered all the comments and suggestions made by the reviewer and the editorial team.

Please find below our point-by-point response to editorial corrections and reviewers’ comments, and a tracked changes version of the revised manuscript. 

Thank you.

Yours Sincerely,

Michael F. Ofori

EDITORIAL COMMENT:

Response: ” Reference # 32” (Aase A, Sandlie I, Norderhaug L, Brekke OH, Michaelsen TE. The extended hinge region of IgG3 is not required for high phagocytic capacity mediated by Fc gamma receptors, but the heavy chains must be disulfide bonded. Eur J Immunol. 1993; 23:1546-51) on page 7 line 156 of the manuscript has been removed because it was a wrong citation. It was supposed to be EBA140RIII-V-{32} and not as Ref (32)

We have also added 4 new references (47-49 and 53) as a result of the recommendation by Reviewer #1 to discuss the results on sub-microscopic parasitaemia

Reviewer #1 Comments

2. The authors adequately responded to all comments. Inclusion of the sub-microscopic data clearly improves the impact of this manuscript but needs to be well discussed.

Just some minor comments

Line 270: 274: Was there any antibody difference between microscopic and submicroscopic groups? Since this data is available it’s should be mentioned as well as discussed for Keta and Hohoe vis a vis Krachi. And how does this affect or influence transmission?, and relate with the authors conclusion that, “These indicate that a threshold of antibody level is needed to control parasitaemia and thus protection”. 

Response: Generally, there was a trend of higher responses in the sub-microscopic group compared to the microscopic group though not significant across the three sites except MSP2-FC27 and Rh2-2030 which were statistically high in the sub-microscopic group in Krachi, we have therefore inserted a statement describing these results under the results section (Page 14 lines 276-279) to reflect the new addition. In addition, the relationship between sub-microscopic parasitaemia and transmission has been discussed (Page 21 line 416-422). Slight modifications have been made on page 21 lines 413 -415.

3. Line 309: MSPDBL_Leucine was not consistently revised correctly. Line 302 vs line 285 reads MSPDBL1_Leucine. Generally, all antigen names should be checked again.

Response: All antigen names have been corrected throughout the text to be consistent ( Eg MSPDBL_Leucine has been changed to MSPDBLLeucine, in Table 2, RH22030 to Pf Rh22030, in Table 2 and throughout the text, etc)

---

## [Decision Letter · Decision Letter 2]

29 Mar 2021

Suitability of IgG responses to multiple Plasmodium falciparum antigens as markers of transmission intensity and pattern

PONE-D-20-38844R2

Dear Dr. Ofori,

We’re pleased to inform you that your manuscript has been judged scientifically suitable for publication and will be formally accepted for publication once it meets all outstanding technical requirements.

Kind regards,

Takafumi Tsuboi

Academic Editor

PLOS ONE

Additional Editor Comments (optional):

Reviewers' comments:

Reviewer's Responses to Questions

**Comments to the Author**

1. If the authors have adequately addressed your comments raised in a previous round of review and you feel that this manuscript is now acceptable for publication, you may indicate that here to bypass the “Comments to the Author” section, enter your conflict of interest statement in the “Confidential to Editor” section, and submit your "Accept" recommendation.

Reviewer #1: All comments have been addressed

2. Is the manuscript technically sound, and do the data support the conclusions?

Reviewer #1: (No Response)

3. Has the statistical analysis been performed appropriately and rigorously? 

Reviewer #1: (No Response)

4. Have the authors made all data underlying the findings in their manuscript fully available?

Reviewer #1: (No Response)

5. Is the manuscript presented in an intelligible fashion and written in standard English?

Reviewer #1: (No Response)

6. Review Comments to the Author

Reviewer #1: (No Response)

7. PLOS authors have the option to publish the peer review history of their article (what does this mean?). If published, this will include your full peer review and any attached files.

Reviewer #1: No

---

## [Editor Report · Acceptance letter]

13 Apr 2021

PONE-D-20-38844R2 

Suitability of IgG responses to multiple *Plasmodium falciparum* antigens as markers of transmission intensity and pattern 

Dear Dr. Ofori:

I'm pleased to inform you that your manuscript has been deemed suitable for publication in PLOS ONE. Congratulations! Your manuscript is now with our production department. 

Kind regards, 

on behalf of

Prof. Takafumi Tsuboi 

Academic Editor

PLOS ONE